# Evidential Data Fusion for Characterization of Pavement Surface Conditions during Winter Using a Multi-Sensor Approach

**DOI:** 10.3390/s21248218

**Published:** 2021-12-08

**Authors:** Issiaka Diaby, Mickaël Germain, Kalifa Goïta

**Affiliations:** Centre d’Applications et de Recherches en Télédétection (CARTEL), Département de Géomatique Appliquée, Université de Sherbrooke, Québec, QC J1K2R1, Canada; Issiaka.Diaby@usherbrooke.ca (I.D.); kalifa.goita@usherbrooke.ca (K.G.)

**Keywords:** pavement surface conditions, multi-sensor systems, data fusion, intelligent systems, deep learning

## Abstract

The role of a service that is dedicated to road weather analysis is to issue forecasts and warnings to users regarding roadway conditions, thereby making it possible to anticipate dangerous traffic conditions, especially during the winter period. It is important to define pavement conditions at all times. In this paper, a new data acquisition approach is proposed that is based upon the analysis and combination of two sensors in real time by nanocomputer. The first sensor is a camera that records images and videos of the road network. The second sensor is a microphone that records the tire–pavement interaction, to characterize each surface’s condition. The two low-cost sensors were fed to different deep learning architectures that are specialized in surface state analysis; the results were combined using an evidential theory-based data fusion approach. This study is a proof of concept, to test an evidential approach for improving classification with deep learning, applied to only two sensors; however, one could very well add more sensors and make the nanocomputers communicate together, to analyze a larger urban environment.

## 1. Introduction

Vehicular mobility can be affected by difficult, or even extreme, weather conditions during the winter period. Indeed, the main causes of accidents are snow-covered or icy roads, which constitute a real source of concern for road management and maintenance services [1]. According to the U.S. Department of Transportation [2], there are more than 5,891,000 vehicle accidents each year. Nearly 21% of these accidents, which represent approximately 1,235,000 cases, are weather-related. In addition, these accidents cause the death of nearly 5000 people on average and are responsible for over 418,000 injuries each year [3]. Most of these crashes are caused by wet pavement (70%) and rain (46%). During winter periods we have a paradoxically low rate of weather-related crashes, with 18% occurring during snow or sleet showers, 13% caused by icy pavement, and 16% occurring on snowy or muddy pavement [3]. Ensuring that roads can be used in a near-normal manner despite the weather conditions, therefore, is a challenge in most Nordic countries, as well as in Canada [4]. For this reason, it is important to put systems in place to monitor the condition of the road network and to forecast future conditions, in order to anticipate maintenance, rehabilitation, and servicing scenarios [5]. This only can be achieved through the acquisition and analysis of reliable real-time data by a road weather service for effective network management [6].

Several sensors are already used to classify the main road surface conditions (dry, wet, snowy or icy), using manual or automated methods. Examples include the non-intrusive DSC111 (spectroscopic) and DST111 (temperature) pavement surface condition sensors that are manufactured by the Finnish company Vaisala [7], Road Condition Monitor RCM511 (optical) by Teconer [8], the IceSight Remote Road Surface Condition Sensor (electro-optical) by High Sierra [9], Starwis (optical) by Lufft/OTT Hydromet [10], and the CCFC Field Camera by Campbell Scientific [11]. Nevertheless, the cost of these sensors is still high (several tens of thousands of dollars), which limits their multiplication in projects with several measurement points.

At the University of Sherbrooke (Quebec, QC, Canada), research has been conducted on this topic, most recently as the subject of a doctoral dissertation [12] characterizing surface conditions by multispectral thermal infrared remote sensing. The principle is based upon measuring the temperature of the pavement at a distance using a moving vehicle. As described by [13], a method of classifying pavement surface conditions has been developed using a dual-channel polarimetric radiometer operating at 94 GHz. This type of sensor, which is based on passive microwave technology, measures the brightness temperature of the pavement surface. The sensors that are presented operate on apparent surface temperature, which does not allow them to detect ice under a film of water, for example.

As presented by [14], a method has been developed for detecting pavement condition using resonant frequency and optical technologies. This type of sensor is intrusive, i.e., it requires direct connection with the pavement. The authors concluded that the capacitance and conductivity of water, ice, and snow could be affected by many factors. For example, de-icing salt has a substantial effect on the dielectric properties of ice and water. In addition, temperature change, ice thickness, and the duration of freezing can cause output drift. Lastly, all of these sensors characterize a single target, the size of which depends upon the detection cone of the device; the latter is the average signal for a circle on the ground surface that is being interpreted.

This paper first demonstrates the potential of multi-sensor fusion or integration using Dempster–Shafer theory [15,16], for the automatic recognition of pavement surface states in real time, while taking into account measurement imperfections. Indeed, recent methods allow combining the Dempster–Shafer theory and CNN classifiers [17,18]. Second, it demonstrates the feasibility of developing a system that is based upon low-cost and high-performance sensors. Last, it confirms the importance of deep learning in aiding automated recognition. Therefore, the method can improve automatic recognition systems for determining pavement conditions during the winter period, by using an intelligent multi-sensor system that is composed of cameras, microphones, and meteorological sensors.

Finally, the innovation of this project lies in the fact that it allows the implementation of a new automatic alert system for the detection of road surface conditions in an artificial intelligence environment that merges information from two distinct classification architectures, one for sound and another for images. In addition, this system is one of the first of its kind in the Quebec climatic context whose goal is to warn in real time road users, as well as the competent authorities, of fast changes to the road surface conditions.

## 2. State of the Art

The monitoring of a road network with sensors is not a new concept. Indeed, as described in [19], a system of detection and classification of symbolic road signs in real time has been designed using the video feed of a camera that is onboard a vehicle. This confirms the potential of video processing for automatic object recognition. Based upon the same principle, in [20], a lane detection approach based upon intelligent vision was implemented. A camera was mounted on the front window of the vehicle to detect lane markings and determine the position of the vehicle relative to the lane lines. Furthermore, in [21], an automated image acquisition and processing system was proposed for the evaluation of pavement surface drainage. For this purpose, a digital camera was integrated into the system to capture images and videos with the desired quality and resolution from the pavement surface, which was subjected to a surface drainage process in adjustable time intervals. We can also take as a reference [22], who developed a real-time weather detection system at a trajectory level capable for providing information on road surface conditions using a single video camera.

Some studies propose using sound-based sensors from which the noise of the tire–pavement interaction is detected for automatic recognition of pavement surface types. For example, reference can be made to [23], in which a method is described for detecting road surface condition according to weather conditions. The tire noise that was emitted by moving vehicles on the road surface by various mechanisms, such as ‘air pumping’ that is caused by air being sucked in, compressed, and released from tire tread [24], was recorded. In [25], a new approach for the identification and mapping of asphalt surface deterioration by measuring tire–pavement interaction noise was also developed. The on-board sound pressure system (OBSP), which is also referred to as the close proximity method (CPX), was applied in a sound pressure level test; a microphone was subsequently installed in a tire–road interaction area. Signal data acquisition was performed over several days and at different times, to account for variability in ambient road conditions, such as temperature, wind speed, and traffic density.

To perform complex perceptual tasks, such as sight and hearing, deep learning is the most suitable tool for implementing intelligent systems, in which the computer can learn and perform tasks almost like a human being [26]. The current most popular learning methods, which offer very good results for supervised or unsupervised classifications, are based upon the use of neural networks [27]. Based upon this observation, a system for recognizing the surface condition of winter roads using a deep convolutional neural network (CNN) that was pre-trained from pavement images was developed by [28]. The data that were used in this prior work had been collected from two sections of highway in southwestern Ontario, Canada, near the unincorporated community of Mount Forest (43°58′54″ N, 80°44′12″ W). The results showed that the use of a pre-trained CNN model is effective in reducing both the need for a large training dataset and long computation times. The model outperformed traditional machine learning models, with an increasing performance advantage, i.e., the quality increased as the quantity of data increased. The CNN model has the further advantage of using raw images without preprocessing, which is required by most traditional systems. In [29], a recurrent neural network architecture was introduced for the automated detection of road surface moisture from tire–pavement interaction noise. The results showed that this method outperformed state-of-the-art support vector machines (SVMs) and achieved outstanding performance on the road moisture detection task, with an accuracy of over 93%. All of these systems, while performing well, use only one sensor for decision-making. They do not handle poor acquisitions, which can be related to various factors: malfunctions, data inaccuracy, and sensor limitations, among others. Therefore, intelligent multi-sensor data fusion is the best way to improve decision-making.

Data fusion is defined by [30] as ‘a process dealing with the association, correlation, and combination of data and information from single and multiple sources to obtain refined position and identity estimates and timely and comprehensive assessments of situations and threats, and their significance.’ Thus, we were particularly interested in the Dempster–Shafer theory (or evidence theory) for data fusion. Indeed, evidence theory is based upon the theory of evidence that was developed by [15] from the earlier work of [16]; hence, the widely used name, the Dempster–Shafer theory (DST). It permits explicit representation of imprecision and uncertainty by assigning a degree of confidence to each hypothesis. According to analyses by [31], this theory has as an essential advantage in the handling of both simple and compound hypotheses, i.e., single and multiple events; thereby allowing great flexibility when modelling various data fusion situations. Moreover, this theory can quantify the conflict between sources.

Applications that are based upon DST have already been proposed, yielding excellent results. In [32], a compelling framework for data fusion in multi-sensor monitoring systems was proposed. It is presented as a unified method to model and fuse detections from various types of sensors that have a priori knowledge of target locations derived from topographical features. They show that the developed belief model provides an effective measure of the data association between tracking and detection. Given the constraints of ongoing system development, the complexity and consistency of the belief function must be controlled by implementing a general recognition framework and limiting the number of focused features. The proof of concept of the proposed data fusion module was achieved by implementing it within the detection system itself. Real-world scenarios have been used to draw conclusions on the localization performance and end-user perception. In [17,18], a new classifier based on Dempster–Shafer (DS) theory and CNN architecture for image classification was presented. In this classifier, the features are then converted into mass functions and aggregated by Dempster’s rule in a DS layer. In [33], DST was used to model and manage uncertainties in applications on the IoT (the Internet of Things, viz., worldwide arrays of sensors and other devices that are connected to the internet and with one another to share data). Such applications are based upon complex event processing. In their approach, the authors studied the identification and management of uncertainties in complex event processing (CEP)-based IoT applications. They proposed DST–CEP, which is a method for using Dempster–Shafer theory to manage uncertainty. Using this theory, their solution combines unreliable sensor data and detects correct results in conflicting situations. In [34], an extended floating car data (XFCD) function was proposed. Based upon this function, vehicles are used as mobile measurement probes for traffic information. The authors showed how to use vehicle data to identify road hazards. The proposed algorithm uses standard vehicle sensor data to classify current road conditions. The fusion of information resulted in a very good detection rate of the weather events that affect road safety.

Our study was, therefore, carried out on the basis of all that has been mentioned above. In the following section we will present the materials and methods that allowed us to carry out our project.

## 3. Materials and Methods

### 3.1. Data

We used a low-cost, high-resolution commercial camera as the optical sensor for color image acquisition, as demonstrated by [35]. The camera used was an Apexcam 4K (Shenzhen Yuqi Technology Co., Ltd., Shenzhen, China) 20 MP, which is a WIFI, underwater waterproof, ultra-HD action camera taking 20 megapixels images. Images were acquired in RGB format. To diversify our data sources, to make our training dataset non-redundant, we used two other data sources. The first was the Canadian Adverse Driving Conditions (CADC) dataset, which is produced by the University of Waterloo (Waterloo, ON, Canada) in partnership with the University of Toronto. These images were acquired using Autonomoose, an autonomous vehicle platform that was created as a joint effort between the Toronto Robotics and AI Laboratory (TRAIL) and the Waterloo Intelligent Systems Engineering Lab (WISE Lab) at the University of Waterloo [36]. CADC includes:56,000 RGB camera images.75 scenes of 50 to 100 images each.10 classes of annotation.Adverse weather driving conditions, including snow.

The second dataset was Rain Fog Snow (RFS), which was published by researchers at the University of Essex (Colchester, UK) and the University of Birmingham (Birmingham, UK) and contains over 3000 images of weather conditions to improve the ability of computer vision models to detect weather conditions [37]. In order to test the reliability of our detection model intended to be tested in a real situation, the images taken by [38] were in the dataset for inference and they are not included in the training dataset. Some examples of these images are shown in Figure 1.

A low-cost microphone was employed as an acoustic sensor for the detection of tire–pavement interaction noise, which was inspired by the method of [39]. The microphone was a VF-M10 3.5 mm cardioid condenser video microphone with unidirectional rifle or interference tube (Viewflex Intelligent Technology Co., Shenzhen, China). This shotgun microphone was coupled to an audio recorder, i.e., a Zoom H1n, which was equipped with a 32 Gb memory card (Zoom Corporation, Tokyo, Japan). The measurement technique that we used was statistic pass-by (SPB), according to ISO 11819-1 (ISO 1997 in [40]). The implementation of this technique is depicted in Figure 2. The second data source, which was made available by researchers from MIT (Massachusetts Institute of Technology, Cambridge, MA, USA), is the ‘MIT wetroad dataset’, which includes dry and wet classes. This dataset is taken from the project entitled ‘Detecting road surface moisture from audio: A deep learning approach’ [29].

### 3.2. Image Classification

In the case of images that were extracted from videos, we proposed a transfer learning solution by testing three models. The objective of the project was to integrate the system on a nanocomputer that was able to analyze the environment in real time. We selected architectures in order to study the performance of a very light architecture (SqueezeNet) versus a heavier and well-known architecture (ResNet50). The three architectures that were tested and compared are: ResNet50, in reference to [28] and described by [41]; MobilenetV3_small, which is an architecture that has been adapted for various mobile platforms (tablets and smart phones, among others) [42]; and SqueezeNet1_1 [43], which is a very small architecture. Table 1 compares the different architectures.

Learning strategy: All default layers were used for each architecture, except for the last two ‘fully connected’ layers of the pre-trained model, which range from 1000 to 4 classes. A model was then generated with the training dataset and the validation dataset.

At the hyperparameter level, we used the stochastic gradient optimization technique, with a value of 0.001 for the learning rate and 0.9 for the momentum. As a loss function, the MSELoss function based on the mean-square error was used.

### 3.3. Audio Classification

The architecture that was selected for the classification of audio signals is M5 [44]. This architecture belongs to the family of classifiers using raw data, unlike the other types that first require extraction of all acoustic features from the audio dataset. This solution aimed to minimize pre-processing of the audio signal and, consequently, avoid slowing down the system on the nanocomputer. There are several similar architectures, depending upon the number of layers that are desired, but our choice was M5 (Figure 3), which has the fastest execution time [44]. The architecture was coded with the Pytorch library.

Learning strategy: The strategy adopted was based upon the tutorial by [45]. The network was fully trained on our dataset. We used the same optimization technique as [45]: an Adam optimizer with a weight de-growth that was set to 0.0001. We trained with a learning rate of 0.01. The loss function that used was NLLLoss, which is negative log-probability loss.

### 3.4. Multi-Sensor Fusion System

The goal of a multi-sensor system is to allow each sensor to fill in the gaps of the other sensors or to allow each sensor to confirm or deny the decisions of the other sensors, which are defined in terms of probability. Data fusion allows a decision to be made based on mathematical reasoning from these probabilities. This fusion guarantees the automation of all tasks and thus allows the system to free itself from any human intervention during the decision process. Ensuring that this decision-making process occurs in real time would allow decision makers to be proactive in guaranteeing the safety of users.

#### 3.4.1. Modelling

For a set of four classes—as the simple assumptions of our project require—there can be as many as 166 elements, viz., simple and compound assumptions (unions and intersections) that need to be defined for a source [46]. Estimating the set of masses, and for each source, is a process that can be very time consuming, especially in the context where we must develop a functional system in real time, and which is integrated with a nanocomputer. In this context, we used a hybrid model that permits the addition of exclusivity constraints, thereby limiting interactions between classes by privileging unions. In other words, we developed an imprecision analysis. Lastly, 15 elements were calculated for each source, i.e., the mass for the four simple classes and the eleven compound classes (class unions).

#### 3.4.2. Estimation

Estimating the set of masses is the most important aspect of data fusion. There are several methods for defining a degree of confidence in the masses of our sources. Indeed, there are almost as many methods as there are authors in the field. For this study, we returned to the basic definition that is specified by Dempster–Shafer evidence theory (DST), i.e., the Shafer consonant method [47]. We adapted this approach for the a posteriori probabilities that are defined in the output of the neural networks; i.e., in the output of the Softmax function that creates probabilities for our four classes.

The mathematical formalism of the consonant method for the initialization of the mass set, i.e., for a source, is defined in the following manner:


*For a set of hypotheses for four simple classes*

$\Theta =\left\{\theta_{1},\;\theta_{2},\theta_{3},\theta_{4}\right\}$




*If the a posteriori probabilities are classified, as follows:*

$P(\theta_{1}|x)>P\left(\theta_{2}|x\right)>P(\theta_{3}|x)>P(\theta_{4}|x)$

*, then*

(1)
$$m\left(\theta_{1}\right)=\frac{P\left(\theta_{1}|x\right)-P\left(\theta_{2}|x\right)}{P(\theta_{1}|x)}\;;\;m\left(\theta_{1}\cup \theta_{2}\right)=\frac{P\left(\theta_{2}|x\right)-P\left(\theta_{3}|x\right)}{P(\theta_{1}|x)}\;m\left(\theta_{1}\cup \theta_{2}\cup \theta_{3}\right)=\frac{P\left(\theta_{3}|x\right)-P\left(\theta_{4}|x\right)}{P(\theta_{1}|x)};\;m\left(\theta_{1}\cup \theta_{2}\cup \theta_{3}\cup \theta_{4}\right)=\frac{P\left(\theta_{4}|x\right)}{P(\theta_{1}|x)}$$



This formalism is used to define the masses (*m*) of the evidential theory for video and audio sources for one class or the union of classes. For this project, Θ represents the set of classes for detection: dry, wet, snowy, and icy. x corresponds to the signal or image to be analyzed by the neural network. $P(\theta_{1}|x)$ is a posteriori probability to get $\theta_{1}$ with x.

#### 3.4.3. Combination

The combination rule in evidential theory can be complex, if one generalizes to all masses, as is the case in the theory of Dezert and Smarandache (DSmT) with the use of non-exclusive classes. In this context, one would need to use a conflict mass redistribution method (known as PCR); there are several that are mentioned in the literature [48]. In the case of a hybrid fusion model with the addition of exclusivity constraints, the combination reverts to the orthogonal combination of Dempster and Shafer, which is defined for two sources as follows:(2)$$m\left(A\right)=\frac{1}{1-Conflict}\underset{B\cap C=A\neq \emptyset }{\sum }m_{1}\left(B\right)m_{2}\left(C\right)$$

And the conflict between the two sources is explained by the following equation:(3)$$Conflict=\underset{B\cap C=\emptyset }{\sum }m_{1}\left(B\right)m_{2}\left(C\right)$$

For Equations (2) and (3), B and C correspond to combinaisons of hypotheses and A=B∩C (e.g., $B=\theta_{1}\cup \;\theta_{3}\;;C=\theta_{1}\cup \;\theta_{2\;};A=B\cap C=\theta_{1}$).

#### 3.4.4. Decisions

Several decision rules are proposed in the literature. Probabilistic mass transformations, such as the pignistic transformation, are very useful in modern multi-sensor tracking systems [40,48,49,50]. Our decision rule is based upon the maximum pignistic probability (a pignistic probability is the probability that a rational individual or agent would assign to an option when required to make a decision [51]).

The pignistic probability is calculated with the following expression:(4)$$P\left(A\right)=\underset{X\in 2^{\Theta }}{\sum }\frac{\left|X\cap A\right|}{\left|X\right|}m\left(X\right)$$
where |.| defines the mathematical expression of the cardinality for a set of elements and ***X*** is a set of hypotheses.

#### 3.4.5. Development

Computer programming of our fusion strategy was realized and optimized in Python, to harmonize it with the neural network coding.

## 4. Results

### 4.1. Image Classification

We created and tested three different models using the ResNet50, SqueezeNet1_1, and MobilenetV3_small architectures, to evaluate their performances against each other and, thus, determine the most suitable model for our project. The training dataset contained 4400 images, of which 80% were selected for training, 15% for validation, and 5% for testing.

The results that are presented in Figure 4 and Table 2 show the evolution of the training process performance and the final performance of the three models. The number of epochs for each model was selected after several simulations (about 10 repetitions). Beyond this threshold, the model was no longer learning. Accuracy was used to evaluate the performance of the model. ResNet50 had the best final performance, with a accuracy of 99.9% for training and 99.1% for validation. Nevertheless, the SqueezeNet1_1 model performed well, despite its very lightweight architecture, and it was the model that was selected later for inference analysis.

We then went on to test the inference for each class, to verify the performance of our model in an environment that had not been used during the training phase. Our test set was composed of 220 images. It included 55 images of each class. The confusion matrix is shown in Figure 5, where the rows represent the observed conditions and the columns represent the predicted conditions. The prediction accuracy was 71% for the observations being correctly classified.

In analyzing the results and verifying our images, we recognized that changes in the environment greatly affected the quality of the results; however, our hypothesis was confirmed that a single sensor is not suitable for monitoring road conditions in a complex environment, and when the training phase is limited to a few well-targeted examples. The image analyses indicated several causes of classification degradation:Presence of both snow and water on the road surface.Presence of tree shade.Large white surface markings were confused with snow.Low ambient light levels.Wet ground with light reflection giving the impression of ice.Road partially covered with snow, water, or ice.Confusion between white ice and snow because of the color.

### 4.2. Audio Classification

The model creation process was carried out on a dataset of 1000 sound files, with 80% for training, 10% for validation, and 10% for testing.

Figure 6 demonstrates the excellent model validation (>99% correction classification). However, problems were encountered in tests within other environments for the ‘ice’ and ‘dry’ classes, where there was 77% and 71% correct classification, respectively. The total accuracy was 87%.

### 4.3. Multisensor Data Fusion

For fusion, we performed a test with the three image classification architectures to validate model choice in our system. The fusion test was performed for two challenging environments: (1) environment #1, with black ice present on the road surface; and (2) environment #2, with melting snow.

#### 4.3.1. Environment #1 (Black Ice)

As a first step, we created a pre-fusion graph and histogram for each model to better understand the effect of sensor merging. Figure 7 shows that the SqueezeNet model defined the surface as either icy or dry, with a very slight tendency related to the wet class, while the audio source shows us that the surface was mostly icy (Figure 7a). We also observe through the fluctuation of the conflict curve that there was, indeed, a greater conflict between the two sources for some images compared to others.

Figure 8 demonstrates that ResNet50 defines the surface as being predominantly icy, as does the audio model (Figure 8a). Both sources are in agreement. In this context, the conflict values are low.

Figure 9 demonstrates that MobileNetV3_small defined the surface as snow, which is a substantial error, while the audio source informed us that the surface was icy (Figure 9a). We also observed from the fluctuations in the conflict curve that there was indeed a major conflict between the two sources.

The sources were subsequently merged, with these results being shown in Figure 10. It can be seen that the decision after fusion, for each of the models, was the presence of ice on the road surface, with a correct classification rate of > 84%. Improvement were made over the single source ‘images’ of both the SqueezeNet model and the MobileNet model. However, fusion degrades the result compared to a Resnet50 model, while preserving the correct class for the final decision. A fusion between the two sources, therefore, is possible and offers good prospects for future applications in this challenging environment, when using lightweight architectures.

#### 4.3.2. Environment #2 (Melting Snow)

Figure 11 reveals that the SqueezeNet model defined the surface as either snowy or icy for the images, but mostly as snowy for the audio files. There was a slight tendency towards the wet class for later audio signals (Figure 11a). Indeed, the level of conflict between the sources increased with acquisition time. This is a normal behavior with the change in surface conditions over time.

Figure 12 shows that the ResNet50 model defined the surface as snow-covered and, thus, made a good classification; as well as the audio presenting the surface as snow-covered (Figure 12a). Conflict values were low between the two sources.

Figure 13 shows that the MobileNetV3_small model defined the surface as predominantly snow-covered and, thus, made a good classification; as well as audio that presented the surface as being snow-covered (Figure 13a). Conflict values were low between the two sources.

We can see that the decision following fusion for each of the models, was a snow-covered surface, with a correct classification rate of >96% (Figure 14). The fusion between the two sources was satisfactory, and this offers good prospects for future applications in this environment.

The final result that is presented is the speed of execution of the systems being implemented. We should not forget that the fusion system must work in real time, while embedded in a nanocomputer. A computational time test was performed on the number of images that could be analyzed in real time; this test was performed on a CPU, viz., an Intel Core i7. Table 3 summarizes the results. The SqueezeNet model could process nearly 4 images per second on a single CPU; considering that a nanocomputer like the Nvidia Jetson Nano is equipped with a GPU (Graphics Processing Unit).

## 5. Discussion

Our project is a proof of concept and has demonstrated that it is possible to implement an automated recognition system for winter road surface conditions, merging image and sound classifications in a deep learning framework. Furthermore, the results illustrate the very promising potential of this system and its efficiency, which are justified for the following reasons: (1) the low cost of the system; (2) light architectures; (3) real-time acquisition and processing; and (4) several possible measurement points.

Indeed, this study represents a major advance in the characterization of pavement surface conditions during winter. Previously, cost problems would not permit the multiplication of measurement points that would yield an abundance of information, and which in turn could improve the accuracy of results. This study solves this problem. Furthermore, this study allowed us to present the performance of combined video and acoustic sensors for the monitoring of pavement surface conditions in a deep learning framework. It also demonstrated the effectiveness and applicability of the DST-based multisensor data fusion method to road weather conditions. Last, with the advent of the Internet of Things and nanocomputers, this study has again confirmed that it is currently possible to develop a real-time and low-cost multi-sensor data acquisition, processing, and analysis system.

As a proof of concept, this study has some limitations. The tests were conducted under specific conditions (daytime), but additional studies should be conducted to test parameters such as seasonality, time of day, night versus day, and sun angle. In addition, this study was made on the basis of the theory of Dempster–Shafer, which has the limitations of non-exhaustiveness, non-exclusivity, and independence of the sources. The theory of Dezert–Smarandache allows passing beyond these limits, and one could, in future studies, develop a system based on this theory. Moreover, as far as the modeling is concerned, 15 elements (union of 4 classes) have been calculated instead of 166 elements (union and intersection). Finally, the estimation was made from the consonant method of Shafer. There is indeed a multitude of methods that could be tested, to choose the most efficient one.

## 6. Conclusions

In conclusion, this study demonstrates that the concept of multi-sensor fusion for automatic detection of pavement surface conditions in the winter period can be brought into being in an artificial intelligence environment using low-cost sensors. Video and audio sensors can be integrated into an embedded and mobile system, to improve detection for better mobility in winter. A multi-sensor system was designed with the use of several neural networks, and the results were merged, based upon evidential theory. The system was optimized to operate in real time and, thus, transmit information about the analyzed environment in an Internet of Things (IoT) context for implementation in smart cities. This automated system would be functional at any time, without requiring human intervention in the decision-making process. Even if the system offers good results in its development phase, additional tests in real situations are necessary to validate the detection of changes in road conditions.

Furthermore, this study used two sensors, but a third sensor, or more, could increase the accuracy of the detection. For example, the weather sensor could take real-time temperatures and provide probabilities from its data that could integrated into the fusion process. Studies are underway to test other sensors and to have several nanocomputers communicate together to classify a larger urban environment.

## Figures and Tables

**Figure 1 sensors-21-08218-f001:**
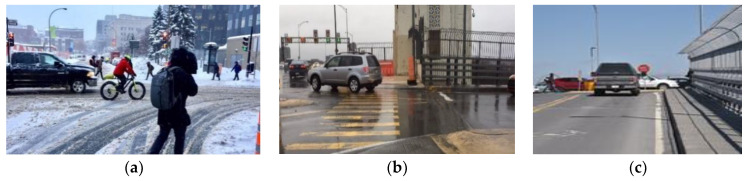
Representation of different types of road surface condition [38]. (**a**) Snowy road; (**b**) wet road; (**c**) dry road.

**Figure 2 sensors-21-08218-f002:**
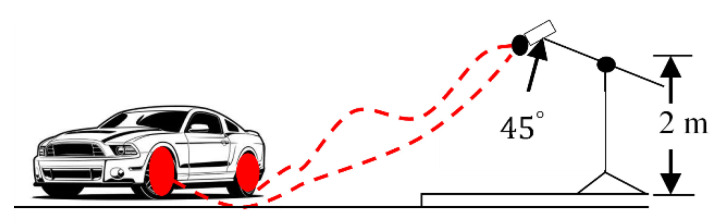
Measurement principle.

**Figure 3 sensors-21-08218-f003:**
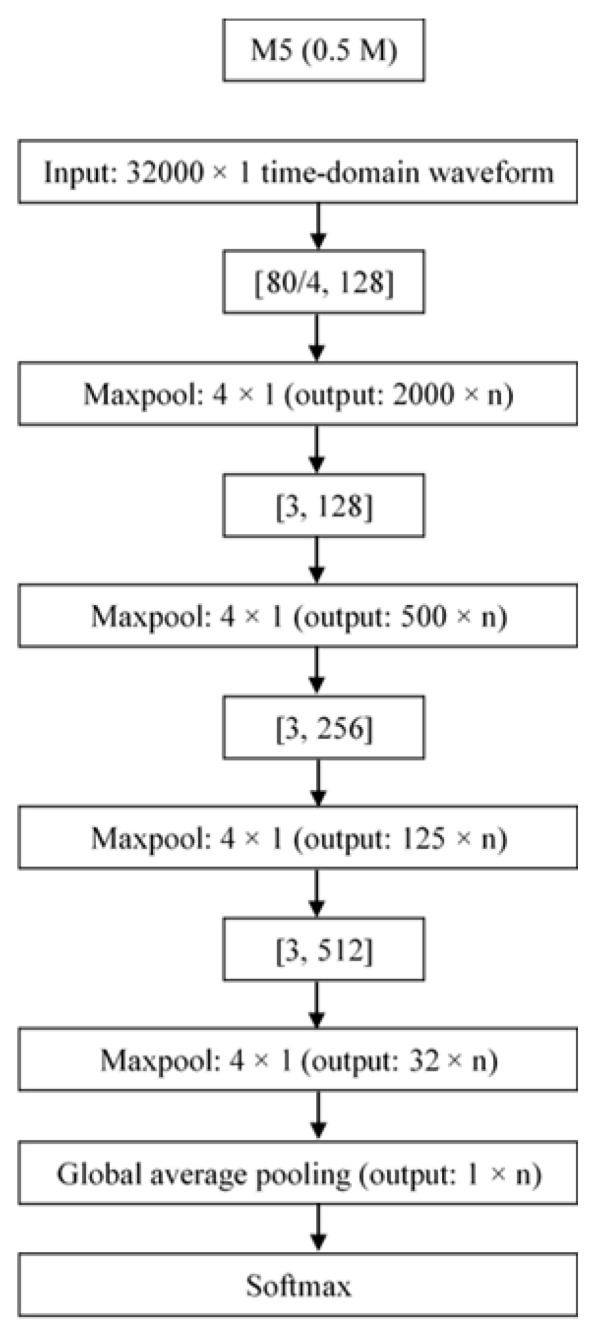
Representation of M5 audio architecture.

**Figure 4 sensors-21-08218-f004:**
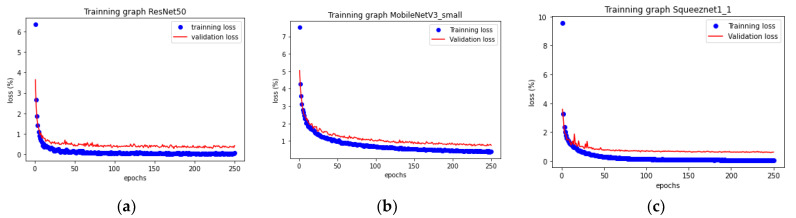
Losses during training: (**a**) ResNet50, (**b**) MobileNetV3-Small, and (**c**) SqueezeNet 1_1.

**Figure 5 sensors-21-08218-f005:**
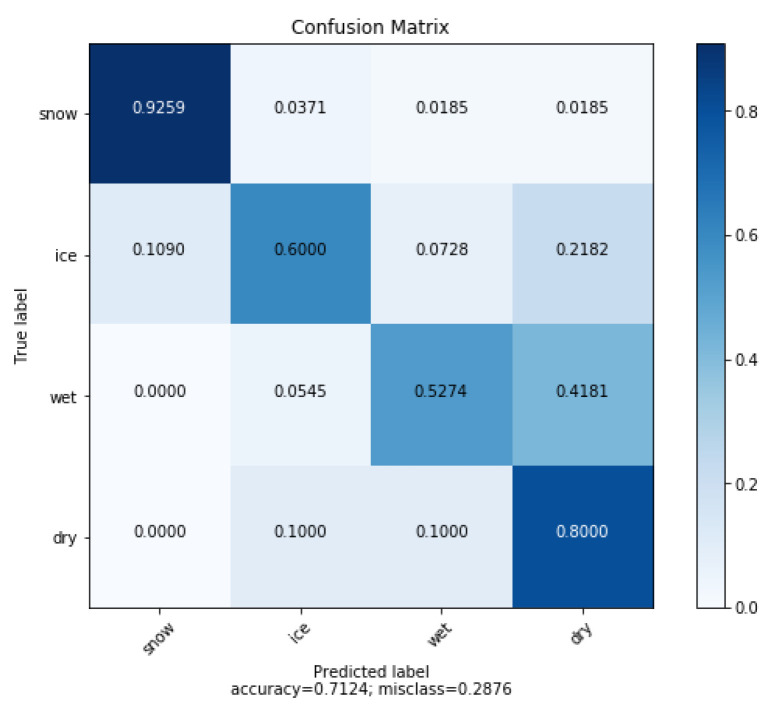
Confusion matrix generated for the test data using the SqueezeNet1_1 model.

**Figure 6 sensors-21-08218-f006:**
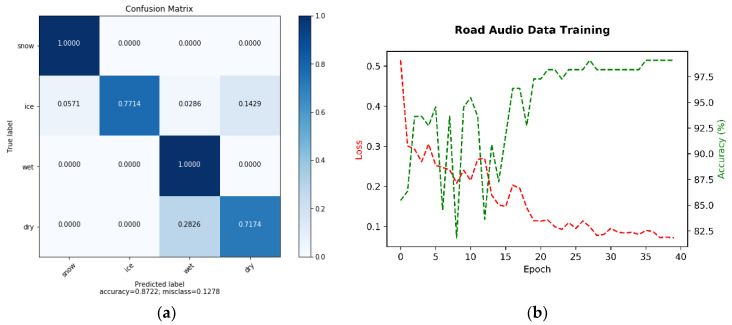
Analysis of the model that was generated for audio classification: (**a**) audio training and validation; (**b**) confusion matrix for the test set.

**Figure 7 sensors-21-08218-f007:**
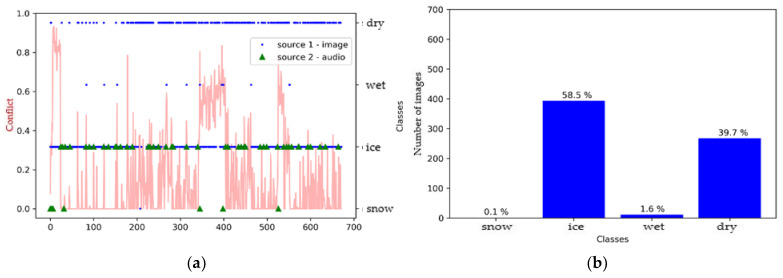
Environment #1. (**a**) Classification of sources with SqueezeNet (video) and M5 (audio) prior to fusion, together with conflict analysis; (**b**) histogram prior to image fusion.

**Figure 8 sensors-21-08218-f008:**
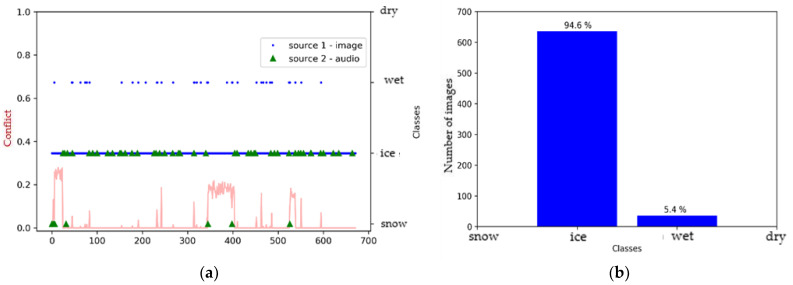
Environment #1 (**a**) Classification of sources with ResnetNet50 (video) and M5 (audio) prior to fusion, together with conflict analysis; (**b**) histogram prior to image fusion.

**Figure 9 sensors-21-08218-f009:**
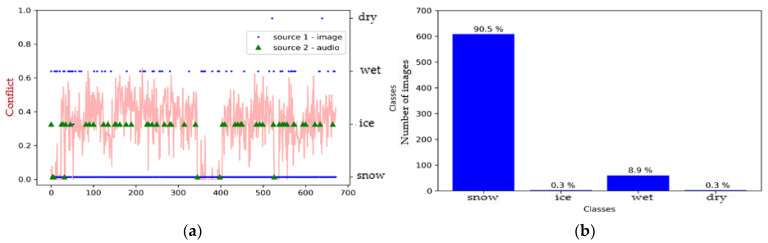
Environment #1. (**a**) Classification of sources with MobileNetV3_small (video) and M5 (audio) prior to fusion, together with conflict analysis; (**b**) histogram prior to image fusion.

**Figure 10 sensors-21-08218-f010:**
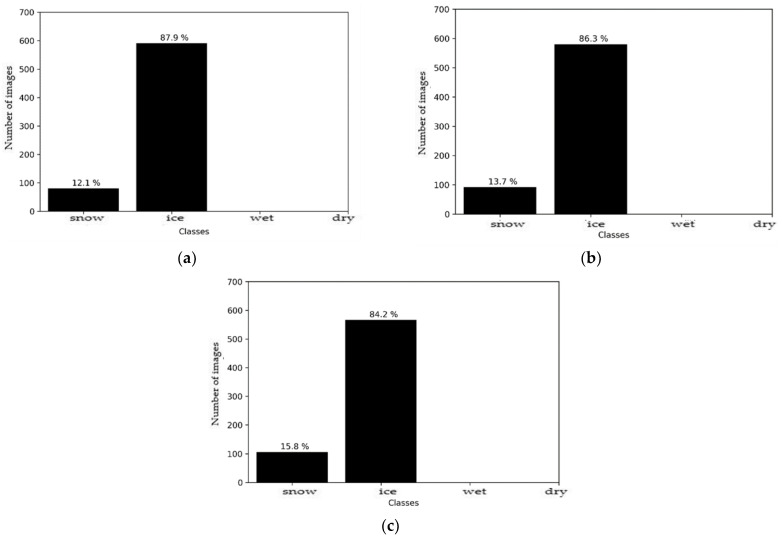
Environment #1. Histogram fusion results with: (**a**) Squeezenet + M5 models; (**b**) ResNet50 + M5 models; (**c**) MobileNetV3_small + M5 models.

**Figure 11 sensors-21-08218-f011:**
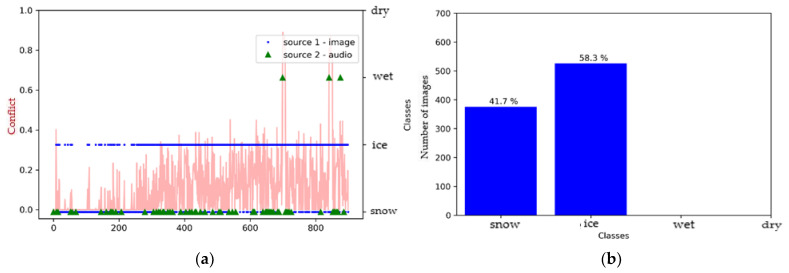
Environment #2. (**a**) Classification of sources with SqueezeNet (video) and M5 (audio) prior to fusion, together with conflict analysis; (**b**) histogram prior to image fusion.

**Figure 12 sensors-21-08218-f012:**
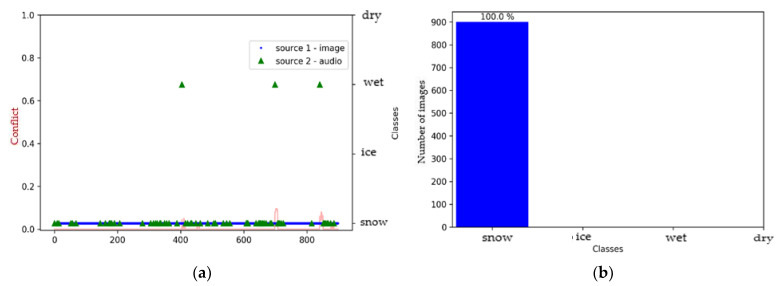
Environment #2. (**a**) Classification of sources with ResNet50 (video) and M5 (audio) prior to fusion, together with conflict analysis; (**b**) histogram prior to image fusion.

**Figure 13 sensors-21-08218-f013:**
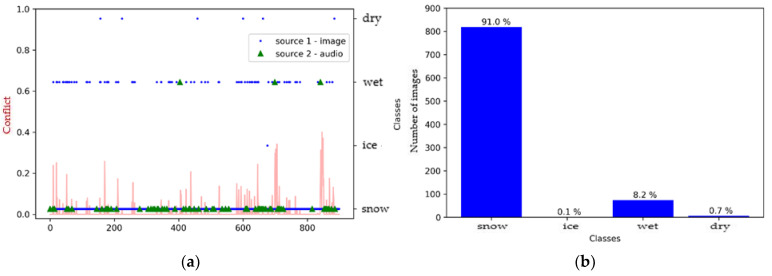
Environment #2. (**a**) Classification of sources with MobileNetV3_small (video) and M5 (audio) prior to fusion, together with conflict analysis; (**b**) histogram prior to image fusion.

**Figure 14 sensors-21-08218-f014:**
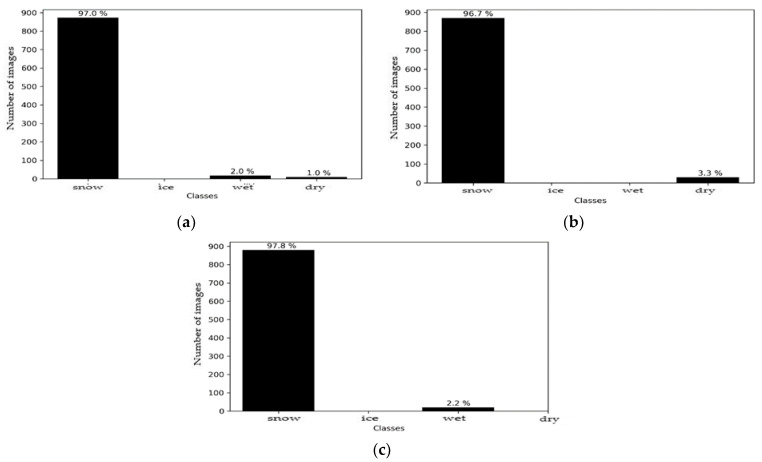
Environment #2. Histogram of fusion results with (**a**) Squeezenet + M5 models; (**b**) ResNet50 + M5 models; (**c**) MobileNetV3_small + M5 models.

**Table 1 sensors-21-08218-t001:** Architectures and parameters of Resnet50 [41]; MobileNetV3_small [42]; and SqueezeNet1_1 [43].

Model	Number of Layers	Number of Parameters
ResNet50	50	25.6 M
MobileNetV3_small	16	1.522 M
SqueezeNet1_1	11	724,548 k

**Table 2 sensors-21-08218-t002:** Performance of the three models.

Architecture	Epoch	Training Accuracy	Validation Accuracy
ResNet50	250	99.9%	99.1%
MobilenetV3_small	100	99.5%	99.0%
Squeezenet1_1	80	99.7%	98.8%

**Table 3 sensors-21-08218-t003:** Video processing speed.

Video Architecture	Duration (s)	Images per Second
Resnet50	347	1.9
Mobilenet	237	2.8
Squeezenet	170	3.9

## Data Availability

Data available on request due to restrictions privacy.

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
