# Peer review of "Evidential Data Fusion for Characterization of Pavement Surface Conditions during Winter Using a Multi-Sensor Approach"

_sensors, 2021, doi:10.3390/s21248218_

Round 1
Reviewer 1 Report
This paper proposed a new data acquisition approach based upon analysis and combination of two sensors in real time by nanocomputer. The first sensor is a camera that records images and videos of the road network. The second sensor is a microphone that records the tire-pavement interaction to characterize each surface condition. The two low-cost sensors are fed to different deep learning architectures that are specialized in surface state analysis; the results were combined by an evidential theory-based data fusion approach. The problem is well described, and the structure of the paper is mostly good. Nevertheless, I have some comments that require revision of the manuscript and the calculations. My detailed comments can be found below.
Q1:The innovation of the research is not explained carefully.
Q2:The introduction should be improved, I noticed some literatures which are old. In this area, The literature should be refered more.
Q3:The Pictures (1,3,9,13) are in poor quality, which needs to be revise.
Q4:Page 1 Line 220-222. The sentence structure is too complicated to understand. It is recommend for 25-30 for word limitation.
Q5:The reason why use Multi-sensor fusion system , please clarify.
Q6:The writers is recommended to consider Artificial intelligence in Engineering data analysis as future prospects part. There are several studies published on the subject of paper. Most of them are not considered in the manuscript. such as:
https://www.sciencedirect.com/science/article/pii/S2352710221004010
Author Response
Thank you for your comments. Please find below the different answers to your questions (the answers have been marked R1,R2.... according to the order of your questions):
R1 (line 79-84) : Thank you for your comment, you are right, we added information. ''The innovation of this project lies in the fact that it allows the implementation of a new automatic alert system for the detection of road surface conditions in an artificial intelligence environment that merges information from two distinct classification architectures, one for sound and the other for images. In addition, this system is one of the first of its kind in the Quebec climatic context whose goal is to warn in real time the road users as well as the competent authorities of the fast changes of the road surface conditions''.
R2 (line 71-72): We added information about sensors, and also about the main topic, i.e. the integration of the evidential theory with deep neural networks
“An evidential classifier based on Dempster-Shafer theory and deep learning” (2021)
https://doi.org/10.1016/j.neucom.2021.03.066
And
“Fusion of Evidential CNN Classifiers for Image Classification” (2021)
https://link.springer.com/chapter/10.1007%2F978-3-030-88601-1_17
R3 : A new figure 1 has been created. We also improved the other figures.
R4 (line 263-266): You are right, we changed the sentence. ''Data fusion allows a decision to be made based on mathematical reasoning from these probabilities. This fusion guarantees the automation of all tasks and thus allows the system to free itself from any human intervention during the decision process.''
R5 : A multi-sensor fusion system allows to take into account the strengths and weaknesses of each sensor for data classification. There are sensors that are efficient at sound, but can generate errors. The same goes for video sensors. The evidential theory allows to take into account the imprecision, the uncertainty and the conflict between sources to improve the decision making on the final classification.
R6 : Thanks for the reference, it's appreciated. The main idea of the article is not to try several machine learning techniques, but it would be interesting to try with several other techniques for another step. The idea is to try to integrate the evidential theory with neural network architectures adapted to sensor classification to improve data analysis and classification.

Reviewer 2 Report
The authors are encouraged to revise the submission according to the attached comments.

Author Response
Thank you for your comments. Please find below the different answers to your questions (the answers have been marked R1, R2.... according to the order of your questions):
R1 (line 18-21): You are right. Thank You. We added a paragraph in the abstract, and more information in the discussion and the conclusion.
Abstract: … “Nevertheless, this study is a proof of concept to test the evidential approach in improving classification with deep learning applied to only two sensors, but one could very well add more sensors and make the nano-computers communicate together to analyze a larger urban environment.”
R2 (line 28-35): Thank you for this reference, it is greatly appreciated. We added information about it.
“According to the U.S. Department of Transportation, there are more than 5,891,000 vehicle accidents each year. Nearly 21% of these accidents, which represent approximately 1,235,000 cases, are weather-related. In addition, these accidents cause the death of nearly 5,000 people on average and are responsible for over 418,000 injuries each year. (Source: 2007-2016 10-year averages analyzed by Booz Allen Hamilton, based on NHTSA data).The vast majority of these crashes are caused by wet pavement (70%) and rain (46%). During winter periods we have a paradoxically low rate of weather-related crashes, with 18% occurring during snow or sleet showers, 13% caused by icy pavement, and 16% occurring on snowy or muddy pavement (Source: 10-year averages from 2007 to 2016 analyzed by Booz Allen Hamilton, based on NHTSA data) “
R3 (Line 45-48) : Thank you. We added new references.
R4 (Line 97-99): Thank you. We added new references.
“We can also take as a reference Khan and Ahmed who have developed a real-time weather detection system at the trajectory level capable of providing information on road surface conditions using a single video camera. “
R5 (line 71-72): It’s done. Thank you.
R6 (120): It’s done. Thank you.
R7(line 174-176): We added a transition as suggested.
“Our study was therefore carried out on the basis of all that has been mentioned above. In the following section we will present the materials and methods that allowed us to carry out our project. “
R8 (line 198): You are right, it’s done. thank you!
R9: You are right, it’s done. thank you!
R10 (line 294-297, line 307, line 319): You are right. We added explanations for each term.
R11: In the literature about deep learning, this is a usual order of magnitude (80% and 20%) to separate training and validation (to analyze model performance and avoid overtraining). For the test, we can use other data or an untrained data set for our model, which is our cas. This is why we use this value scale. As we have less sound data, the percentage is a bit larger for the test.
R12: We changes the scales with your recommendations.
R13 (line 337): You are right. We tested about ten times for each model (image and audio). We added this order of magnitude.
R14 (line 474-476): This information is relevant to the reader. We have added the information in the limitations of this study (Discussion). We have mainly focused our study on the feasibility of combining sensor analyses by deep learning and using the evidential theory (Dempster-Shafer).
“As a proof of concept, this study has some limitations. The tests were conducted under specific conditions (daytime), but additional studies should be conducted to test parameters such as seasonality, time of day, night versus day, and sun angle.”
R15: You are right, it’s done. thank you!
R16: You are right, it’s done. thank you!
R17(line 474-484): we added one paragraph about limitations
“As a proof of concept, this study has some limitations. The tests were conducted under specific conditions (daytime), but additional studies should be conducted to test parameters such as seasonality, time of day, night versus day, and sun angle. Also, this study was made on the basis of the theory of Dempster-Shafer which presents limits of non-exhaustiveness, non-exclusivity and independence of the sources. The theory of Dezert-Smarandache allowing to pass beyond these limits, one could for the future studies, develop the system based on this theory. Moreover, as far as the modeling is concerned, 15 elements (union of 4 classes) have been calculated instead of 166 elements (union and intersection). Finally, the estimation has been made from the consonant method of Shafer. There is indeed a multitude of methods that could be tested to choose the most efficient one.”
R18 (line 498-502): yes, thank you. We added a paragraph.
“Nevertheless, this study used two sensors, but a third or more sensor could increase the accuracy of the detection. For example, the weather sensor could take real-time temperatures and provide probabilities from its data that will be integrated into the fusion process. Studies are underway to test other sensors and to have several nano-computers communicate together to classify a larger urban environment“.

Round 2
Reviewer 2 Report
The reviewer thanks the authors for their detailed attention to the first round of reviewer comments. The reviewer has no further comments to improve the manuscript.